# Assessment of Stress Levels in Lactating Cattle: Analyzing Cortisol Residues in Commercial Milk Products in Relation to the Temperature-Humidity Index

**DOI:** 10.3390/ani13152407

**Published:** 2023-07-25

**Authors:** Mohammad Ataallahi, Si Nae Cheon, Geun-Woo Park, Eska Nugrahaeningtyas, Jung Hwan Jeon, Kyu-Hyun Park

**Affiliations:** 1Department of Animal Industry Convergence, Kangwon National University, Chuncheon 24341, Republic of Korea; ataallahim@kangwon.ac.kr (M.A.); rmsdn92@kangwon.ac.kr (G.-W.P.); eskanugrahaeningtyas@kangwon.ac.kr (E.N.); 2Animal Welfare Research Team, National Institute of Animal Science, Rural Development Agriculture, Wanju 55365, Republic of Korea; wns2546@naver.com (S.N.C.); jeon75@korea.kr (J.H.J.)

**Keywords:** commercial dairy milk product, cortisol residue, heat stress, milk stress marker, temperature-humidity index

## Abstract

**Simple Summary:**

Cortisol is not a component of milk production; hence, it is generally not expected to be present in milk in substantial proportions. However, in some cases, cortisol may be present in milk due to stressful conditions or other factors such as the use of steroid medicines. On the other hand, cortisol as a steroid hormone is generally considered to be thermally stable. Therefore, heat processing such as sterilization and pasteurization methods may not significantly affect the amount of cortisol. This study evaluated the relationship between the milk cortisol concentration in commercial milk products and the temperature-humidity index at the time of milk production. The results indicated that a higher milk cortisol concentration was released in high temperature and humidity environments. The higher average cortisol in July may be attributed to the hotter and more stressful environmental conditions present during this month. Evaluating the milk cortisol concentration in commercial milk products may help in improving the quality of dairy farming practices and its potential application in estimating the impact of environmental stressors on lactating cattle.

**Abstract:**

Chronic stress in the dairy cattle industry has negative impacts on animal health, productivity, and welfare. It has been confirmed that cortisol transfers to milk and resists the high temperature during milk processing. This study evaluated the relationship between the milk cortisol concentration (MCC) in commercial milk products and the temperature-humidity index (THI) at the time of milk production. Eleven commercially produced pasteurized and sterilized milk products, purchased in Chuncheon, Korea, with production dates ranging from July to October 2021 were analyzed. The MCC was extracted using diethyl ether and analyzed using an enzyme immunoassay. The average THI values based on microclimate data provided by the Korea Meteorological Administration were 77 ± 0.8, 75 ± 1.4, 69 ± 1.4, and 58 ± 1.8, in July, August, September, and October, respectively. The average MCC levels were 211.9 ± 95.1, 173.5 ± 63.8, 109.6 ± 53.2, and 106.7 ± 33.7 pg/mL in July, August, September, and October, respectively. The MCC in July was higher than in August, September, and October (*p* < 0.05), while it was lower in September and October than in August (*p* < 0.05). Significant variations in the MCC were observed in commercial milk products across the four production months (*p* < 0.05), except for two milk products. Overall, monitoring the cortisol residue in commercial dairy milk products can be an alternative indicator of stress in dairy cattle of farms.

## 1. Introduction

Environmental changes may have significant impacts on the dairy industry as it is a large and important sector of the agriculture industry. For example, high temperatures can cause heat stress in cattle, reducing their appetite, milk production, reproductive performance, increasing the risk of health problems, and reducing welfare [1]. Heat stress causes cortisol to be released systemically into the bloodstream of cattle [2]. This systemic release of cortisol is a physiological response that helps the cattle’s body cope with the heat stress [2].

Blood flow is crucial for milk production in dairy cattle and approximately five hundred liters of blood must flow through the udder of a healthy dairy cow to produce just one liter of milk [3,4]. However, milk production is not solely dependent on blood flow. It can be influenced by various factors, including genetics, nutrition, and management practices [1,5]. Milk as a biomatrix contains a variety of essential components such as proteins, fats, carbohydrates, vitamins, and minerals [1]. Moreover, dairy milk is ranked as a high quality food source because of a well-balanced composition of essential nutrients and it is an easily digestible form for all adult consumers [6,7], which is considered one of its most significant characteristics.

Assessing the stress levels in lactating cattle is an important tool for monitoring welfare, identifying potential sources of stress, and developing management strategies for mitigating stress and improving food safety [8]. The measurement of analytes that mediate the physiological stress response is a way to monitor the effects of environmental stressors. Traditionally, stress assessments in cattle have been carried out using hormonal analyses through blood sampling. However, blood sampling is a stressful procedure and it disturbs the cattle, causing additional stress [9]. Therefore, less stressful procedures have been developed in various liquid and non-liquids biomatrices such as saliva, urine, milk, feces, and hair [2,10]. Changes in cortisol levels in the bloodstream can affect the milk cortisol concentration (MCC). An abnormal MCC has been linked to cattle stress or health problems [11,12], as well as factors such as the cattle’s reactions to their environment, handling techniques, feeding practices, housing conditions, and milking procedures. The environment in which cattle are housed may have a significant impact on their stress levels and, subsequently, their MCC. In addition, extremes in temperature, poor ventilation, crowding, and exposure to loud noise can all cause stress reactions in cattle and rise their MCC. Moreover, changes in the diet’s nutritional content or feeding schedules can also impact the MCC and cow hormonal balance. In order to keep the MCC at its ideal level and reduce the stress level in dairy cattle, monitoring and regulating these parameters are crucial. On the other hand, cortisol as a steroid hormone is generally considered to be thermally stable. Consequently, processing methods such as sterilization and pasteurization may not significantly affect the MCC in dairy cattle [13,14] and humans [15,16]. However, heat treatment as a necessary procedure to ensure product safety and extend shelf life can destroy both harmful and beneficial microorganisms, which can result in losses of nutrients in dairy and human milk products [17,18,19].

The combined effects of high temperature and humidity as environmental factors may exacerbate stress in high-producing dairy cattle, resulting in a high cortisol concentration in milk. Thus, we hypothesized that for a high temperature-humidity index (THI), the level of MCC will also be high. Heat stress occurs when an animal’s heat load is greater than its ability to dissipate heat. Therefore, this study was designed to assess the relationship between the MCC in commercial dairy milk products and the THI at the time of milk production. Measuring cortisol residue in commercial dairy milk products is a non-invasive and possibly a convenient method for monitoring the stress levels of lactating cattle. Additionally, it does not require handling or sampling from the cattle. Commercial dairy milk products are easily accessible and readily available, making them a practical option for monitoring cattle welfare.

## 2. Materials and Methods

### 2.1. Commercial Milk Collection and Preparation

In total, 11 products of commercially available dairy milk were selected for analysis in this study. These products were chosen from among the most frequently consumed milk brands in Chuncheon city, Korea. Milk of various types was purchased weekly from July to October 2021 as shown in Table 1. All milk products were made from 100% raw milk. From each milk product, 50 mL was sampled into a conical tube and stored at −20 °C for subsequent analysis. The milk samples were analyzed prior to the expiration date, and no additional preparation steps, such as centrifugation, were required. Though individual cow or farm information (e.g., number of milking cows, their management condition, and the intrinsic and extrinsic parameters) may influence the MCC, these factors were attenuated as commercial milk is collected from a small number of farms in the same area and mixed altogether; thus, the possible significant factor for the release of the MCC is the environment, particularly the THI in this study.

### 2.2. Milk Cortisol Analysis via Enzyme Immunoassay

The measurement of the MCC was based upon a method described by Fukasawa et al. [20] on Holstein milk. For cortisol extraction, the milk samples were prepared by thawing in a 37 °C water bath and 0.1 mL of milk was mixed with 0.9 mL diethyl ether under a fume hood (Figure 1). The mixture was vortexed for 1 min, then two layers, organic (ether phase) and inorganic (aqueous phase), were made. The organic layer containing cortisol was transferred into a separate 2 mL micro-centrifuge tube and evaporated under a laboratory fume hood over 2 h. The extracted residues were stored at −20 °C for further analysis. It should be noted that multiple extractions (twice or three times) are sufficient for the full extraction of cortisol and the amount of evaporated organic layers for the final calculation should be measured. The dried evaporated residue was re-dissolved in 0.25 mL of assay buffer supplied with the enzyme immunoassay (ELISA) kit, (Enzo Life Science, Farmingdale, NY, USA) at room temperature, vortexing for 1 min, and 0.1 mL of the reconstituted sample was assayed in duplicate into a 96-well plate. The kit contains a 96-well plate format and allows 37 samples to be analyzed in duplicate. This is because up to 22 wells are required for the calibration of assay standards and so that the controls for the conformation of the assay work correctly. The ELISA kit sensitivity was 56.72 pg/mL. Milk samples need to be diluted with a steroid displacement reagent (SDR) at a ratio of 1:100 (1 part of SDR for every 99 parts of milk samples). The SDR inhibits steroid binding to proteins, allowing it to be detected by the assay. The optical density of the samples was read using a microplate reader (SpectraMax absorbance readers, Molecular Device LLC, San Jose, CA, USA) at a wavelength of 405 nm. The concentration of cortisol in the milk extracted was calculated by a four-parameter logistic calibration (4PLC) curve fitting program provided in the microplate reader. To ensure the accuracy and reliability of the results obtained, it is generally recommended that the percent coefficient of variation (CV%) within the plates (intra-assay) should not exceed 10–15%, and the variation between the plates (inter-assay) should not exceed 15–20%, as per the guidelines provided by the ELISA assay manufacturer’s protocols. The average CV% was calculated as 15.4% for the intra-assay and 19.8% for the inter-assay in current study.

### 2.3. Temperature-Humidity Index

The average monthly temperature and relative humidity data for the period from July to October 2021 were obtained from the website of the Korea meteorological administration (KMA), https://data.kma.go.kr/cmmn/main.do (accessed on 28 February 2023). These data were obtained from 81 KMA stations located in close proximity to the cattle farming sites across the country. The index of THI was calculated using a previous study described by Ataallahi et al. [21], with higher THI values indicating a greater risk of heat stress for cattle [22]. The THI values greater than 72 indicate the onset of heat stress, with levels between 72 and 79 considered mild stress, 80 to 89 considered moderate stress, 90 to 98 considered severe stress, and levels above 98 considered dangerous [23,24].

The data obtained from the KMA stations provided valuable information on the local climate conditions, enabling the calculation of THI values and subsequent evaluation of heat stress risks for cattle. The average THI levels calculated monthly from 81 weather stations were 77 ± 0.8, 75 ± 1.4, 69 ± 1.4, and 58 ± 1.8 in July, August, September, and October, respectively (Figure 2).

### 2.4. Statistical Data Analyses

The statistical analysis was performed using the SAS software (version 9.4, SAS Institute Inc., Cary, NC, USA). Under the general linear model (GLM) procedure, Tukey’s studentized range (HSD) test was used to determine the difference between the MCC and the production months of commercial milk samples. In this experiment, the commercial milk products were considered as a random effect. The THI from each month of milk production was treated as a fixed effect. All results are presented as mean ± standard deviation (SD) and statistical significance was considered at *p* < 0.05.

## 3. Results

### Cortisol Concentration in Commercial Dairy Milk

In total, 11 products of commercial dairy milk were analyzed for MCCs, with results indicating average levels of 211.9 ± 95.1, 173.5 ± 63.8, 109.6 ± 53.2, and 106.7 ± 33.7 pg/mL for the months of July, August, September, and October, respectively (Figure 3). The average MCC in July was significantly higher (*p* < 0.05) than in August, September, and October, while the average MCC in September and October was significantly lower (*p* < 0.05) than in August (Figure 3). However, there was no significant difference (*p* > 0.05) between the average MCC in September and October (Figure 3).

The MCC may be influenced by heating and holding time during the processing of milk products. The MCC from each of the commercial milk products at four production months are shown in Figure 4 and significant differences were observed within them (*p* < 0.05). However, the average MCC was not different significantly (*p* > 0.05) with the months of milk production in commercial milk product numbers (4) and (11) (Figure 4). The effect of THI on the MCC was calculated based on the weather condition area of the milk production for each commercial product.

## 4. Discussion

This study analyzed cortisol residue levels in commercial dairy milk products produced in Korea. Cortisol is a natural hormone produced by the adrenal glands and is released in stressful conditions and at a low blood-glucose concentration [10]. There is a positive correlation between stress level and blood cortisol concentration, which can also be found in cattle milk [12,25]. In Korea, lactating dairy cattle produce milk year-round, although the milk production and composition may vary during different seasons [12,24]. In the current study, all commercial milk products included in this analysis were derived from 100% raw milk, ensuring that the samples used for the cortisol residue measurement represented milk in its natural, unprocessed form. In addition, this approach provided a comprehensive understanding of cortisol, without the influence of additional processing steps or additives that could potentially alter cortisol concentrations. Previous research by Ito et al. [26] reported an MCC ranging from about 500 to 10,000 pg/mL and mentioned that the level of MCC was lower than the cortisol level in blood samples. However, the average MCC levels observed in commercial milk samples in the present study were less than the previously reported range [26,27]. The results indicate that cortisol can still be detected in commercial dairy milk products even after sterilization and pasteurization methods, and its concentration and structure are not significantly affected during heat processing [13,14]. Gellrich et al. [27] mentioned that single blood samples do not provide sufficient information as cortisol is released in a pulsatile manner with typical pulse intervals of about 120 min. Therefore, cortisol in milk might reveal information about the mean concentrations in the bloodstream [27]. The analysis of 11 commercial dairy milk products revealed varying MCCs in different hot months. Currently, there are no established limits for the amount of cortisol residues in milk or other dairy products in Korea. However, other countries have set maximum residue limits for cortisol in dairy products. For instance, Japan has set a maximum residue limit of 10 µg/kg for cortisol [28]. Analyzing the cortisol residue in commercial milk may reveal the sources of cortisol secretion. Heat stress causes physiological responses in dairy cattle through increasing cortisol levels by activating the hypothalamic-pituitary-adrenal axis (HPA), which regulates cortisol secretion [10,27]. Moreover, in both acute and chronic stress situations, multiple stressors (physical or psychological) are caused which increase cortisol [10]. Environmental stressors such as high temperatures and humidity can affect cattle by reducing their feed intake and lowering the milk yield, as well as changes in the milk composition [1,29]. In the hot environmental months, lactating cattle were more likely to experience heat stress, which affected the MCC [24]. Heat stress can cause physiological and biochemical changes in dairy cattle, which can affect both milk production and composition [30,31,32]. Heat stress can also cause health problems in cattle, such as reduced fertility and an increased susceptibility to diseases [8]. The animals’ ability to adjust to the hot environment after a few weeks may be the cause of the greater MCC in July than in August despite the THI levels of both months being of mild heat stress (THI over 72; Figure 3); this is how the body’s homeostasis is maintained [33,34]. Therefore, the MCC was observed to be higher in July due to the increased environmental THI, which reached up to 77 levels. This suggests that lactating cattle during the month of July experienced higher levels of stress, leading to an elevated MCC. Cortisol can help cattle to cope with heat stress, as the cattle are sensitive to changes in ambient temperature and humidity. The optimal temperature range for cattle is between 10 °C and 15 °C [35]. The MCC decreased in the months following July due to a decrease in THI levels, indicating a decrease in stress levels during this period. However, no significant difference was observed between the average MCC in September and October, suggesting a relatively stable stress level during these two months. The variation in the MCC across different months can also be attributed to several factors. It is well-established that environmental conditions, such as temperature and humidity, play a crucial role in determining the stress levels experienced by cattle. The higher average MCC in July may be attributed to the hotter and more stressful environmental conditions present during that month. Heat stress, induced by high temperatures and humidity, can lead to physiological and biochemical changes in cattle [36], affecting their cortisol production and subsequently influencing the MCC. As the summer months progress, cattle may gradually adapt to the environmental stress, resulting in lower cortisol levels and decreased MCCs in subsequent months. Furthermore, other factors, such as management practices, breed characteristics, and individual variation among cattle, could also contribute to the observed differences in MCCs.

In Korea, weather conditions might affect milk production and composition in highly productive lactating cattle [24]. Based on the THI levels, the weather condition was in mild stress in July, which could potentially impact cortisol as a very sensitive indicator of heat stress [37]. However, measuring the residue of MCC can be challenging due to various factors, such as breed [12], handling, and transportation procedures [38], which can affect the MCC. Furthermore, the MCC can vary depending on the time of day, with higher levels in the blood in the morning than in the evening [25]. However, there is a lack of information regarding the milk collection process for each commercial dairy production and the types of lactating cattle farming systems, such as whether they are indoors or outdoors. During hot months, cattle may not have access to enough pasture, which could result in reduced milk production. Additionally, cattle require access to sufficient water to stay hydrated and maintain milk production during hot weather [5,24]. If the cattle do not have access to enough water, their milk production may decrease, and their MCC may be affected [39]. The influence of heating and holding time during milk product processing on MCCs were not examined in the present study. The MCC from each commercial milk product across four production months was compared and significant differences were observed within the production months except in the commercially processed milk product numbers 4 and 11, where the average MCC did not differ significantly among the production months. The processing steps, such as pasteurization or sterilization, are necessary for ensuring the safety and extending the shelf life of dairy products [14]. It is possible that the processing conditions employed in these products were optimized to minimize cortisol degradation or so that the heat treatment did not affect the stability of the cortisol significantly. The analysis of MCCs in most of the commercial milk products revealed that the presence of cortisol residue was high in milk products that were collected in July and August. This is perhaps because of the effect of the THI on dairy cattle. A comparison of THI values indicated that the cattle may have experienced mild heat stress in July and August than in September and October. Furthermore, milk produced during the summer months tends to have higher levels of fat and protein compared to milk produced in the winter, which is attributed to the higher quality of feed during the summer [5,39]. As a result, differences in lactating dairy milk produced during months with high THI values compared to other months with normal THI values could be a factor of heat stress and increase of cortisol. In the current study, microclimate data provided by KMA were used to calculate the THI level. Based on the findings, variations in the THI levels across four different months were observed. In July, the average THI was high, followed by a slight decrease in August. September exhibited a further decline, while the THI value in October recorded the lowest in Korea. The fluctuation in the THI levels may be attributed to the seasonal changes in Korea. The highest THI level was recorded in July, due to July being in the midst of summer. This can be attributed to the combination of high temperatures and humidity during this time, which are typical characteristics of the summer season in Korea. As the summer season progressed, there was a gradual decrease in THI levels, indicating a transition towards more moderate thermal conditions. September showed a further decline in the THI compared to August as the autumn season begins in September and temperatures generally become milder. Finally, October recorded the lowest THI value, indicating a significant reduction in temperature. This can be attributed to the arrival of autumn, which is characterized by cooler temperature conditions in Korea. The results of the THI calculation in this study provide the seasonal fluctuations in environmental stressors. The THI reflects the combination of temperature and humidity levels, particularly during the summer months, indicating a greater risk of heat stress for dairy cattle.

This study possesses both strengths and limitations that should be taken into account in future studies. To the best of our knowledge, this is the first study to analyze cortisol residue in commercial milk products specifically in Korea. This study focused on the milk products most commonly consumed in Korea; it provides valuable information related to the level of cortisol residue in milk food products. However, it is important to acknowledge the limitations of the study. One of the primary limitations is the lack of existing research on the measurement of cortisol residue in dairy milk products in Korea, which limited our interpretation of the findings. Furthermore, the study was limited by the number of samples analyzed or the specific brands included. A larger sample size and inclusion of a wider range of brands with more information about milk collection methods, or storage methods, can provide a more comprehensive understanding of the cortisol residue in commercial milk products in Korea. The level of cortisol is influenced by not only extreme environmental temperature and humidity, but also other parameters such as feeding practices, housing systems, and methods of milk harvest. For instance, Fazio et al. [11] explained that the activation of the HPA axis does not occur in response to machine milking. However, they confirmed that the increase in cortisol levels during milking under normal conditions appears to be regulated by mechanisms other than stress. Additionally, the available analytical techniques should be evaluated. A variety of instrumental analyses is designed to measure the cortisol present in liquid and non-liquid biomatrices. Gas chromatography–mass spectrometry, liquid chromatography–mass spectrometry, high-performance liquid chromatography, and ELISA are some of the methods commonly used [2,10]. Each of the aforementioned methods has its advantages and limitations, and the choice of method depends on factors such as sensitivity requirements, sample type, cost, and simplicity. Due to the low concentration of cortisol residue in the milk matrix, it needs to be extracted efficiently for an accurate determination of the total cortisol before analysis and detection. Extraction eliminates potential interfering substances. Therefore, it is essential to use appropriate sample pre-treatment methods. Currently, one of the reported extraction approaches for cortisol in milk is liquid–liquid extraction (LLE) using diethyl ether or other organic extraction solvents. In this experiment, positive and negative matrix controls were not performed during cortisol extraction from commercial milk samples. This limitation should be taken into account for extraction efficiency. Researchers need to carefully consider these factors when selecting an appropriate method for cortisol analysis in milk. The presence of cortisol residue in commercial milk products is not a health concern for consumers, as the levels are typically very low and harmless [28]. However, it could be a quality control issue for the dairy industry and may affect the taste and quality of the milk. Thus, screening the cortisol residue in commercial dairy milk products is important for maintaining product quality and ensuring consumer satisfaction. It is suggested that further research should investigate the risk and side effects of consuming dairy products that exceed daily body requirements and possible exposure to very small levels of cortisol residue in commercial dairy milk products. Understanding the potential implications of consuming milk with higher cortisol residues is essential for ensuring consumer health and safety. Furthermore, the dairy industry can continue to provide high-quality milk products, while ensuring consumer satisfaction and promoting dairy cattle welfare.

## 5. Conclusions

This study contributes to the development of less stressful techniques for assessing stress in dairy cattle and informs management practices for improving animal welfare and product quality. Monitoring cortisol residues in commercial dairy milk products could be a useful method for identifying the sources of stress in lactating cattle on farms. Additionally, it could serve as a valuable parameter for evaluating the quality of milk and dairy products on the market. However, it is important to note that the milk cortisol concentration should be used in combination with other welfare indicators and interpreted with care.

## Figures and Tables

**Figure 1 animals-13-02407-f001:**
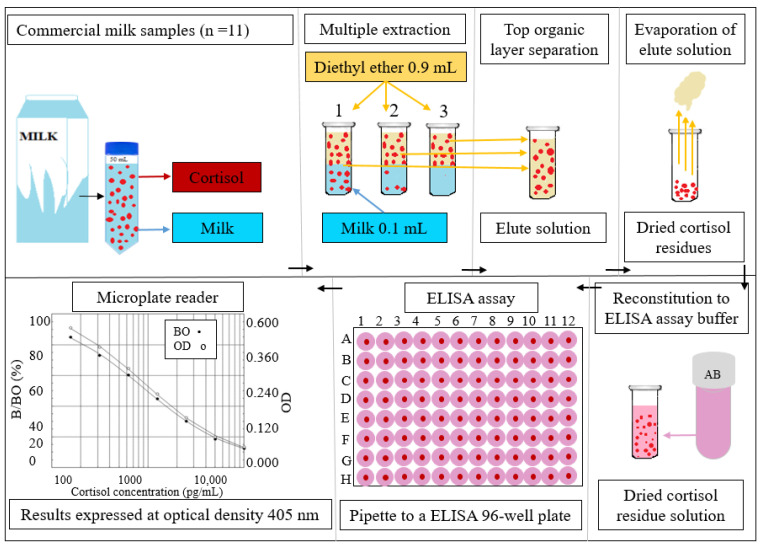
The procedural summary of commercial milk sample preparation and analysis via ELISA kit assay.

**Figure 2 animals-13-02407-f002:**
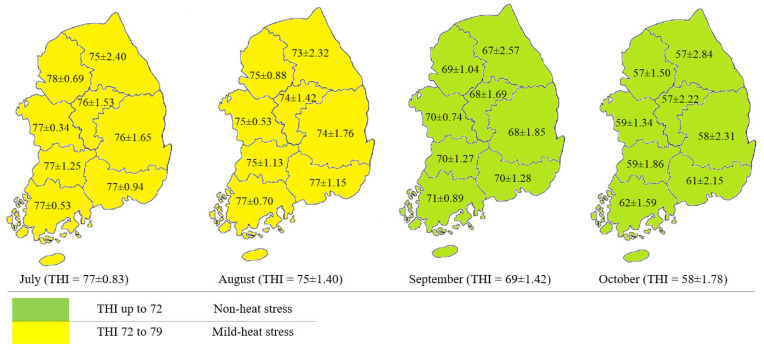
Shows the temperature-humidity index (THI) across Korea from July to October 2021. During milk production, lactating cattle experienced mild heat stress levels (THI between 72 and 79) in July and August and normal conditions (THI under 72) in September and October.

**Figure 3 animals-13-02407-f003:**
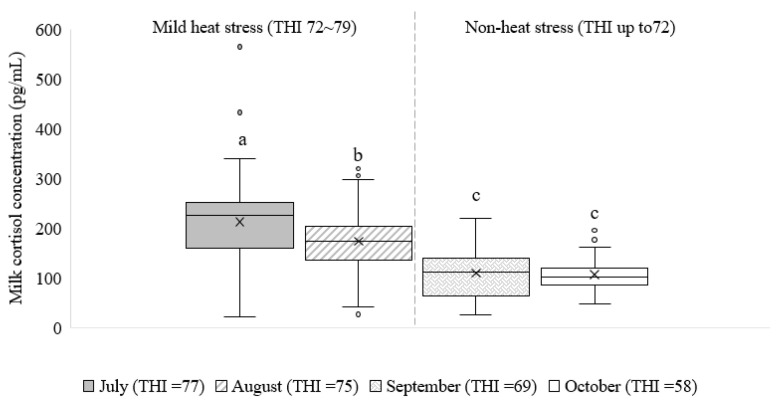
Boxplot showing the milk cortisol concentration (MCC) in total commercial milk products across four months of milk production. The average temperature-humidity index (THI) for each month of milk production was calculated as a key relationship determining the MCC. Error bars represent the range of minimum and maximum cortisol values observed in the milk samples. The outliers are shown as white circles and the mean are indicated by the x. Different letters indicate significant differences (*p* < 0.05).

**Figure 4 animals-13-02407-f004:**
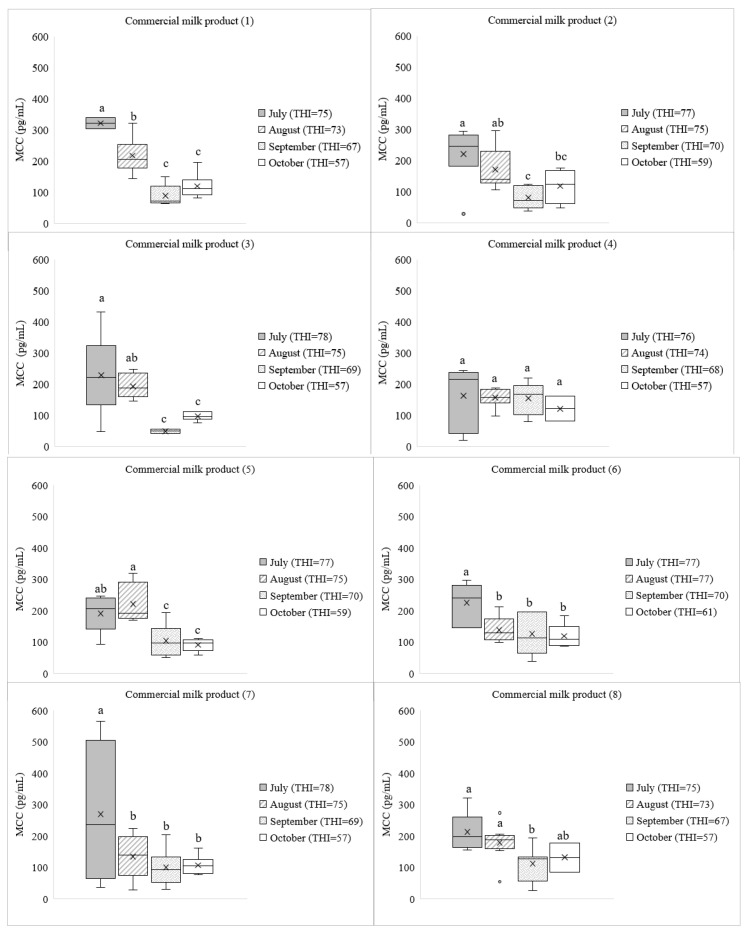
Boxplot showing the milk cortisol concentration (MCC) from 11 commercial milk products at four months of milk production. The average temperature-humidity index (THI) for each month of milk production was calculated as a key relationship determining the MCC. Error bars represent the range of minimum and maximum cortisol values observed in the milk samples. The outliers are shown as white circles and the mean are indicated by the x. Different letters indicate significant differences (*p* < 0.05).

**Table 1 animals-13-02407-t001:** Summary of 11 products of the commercial dairy milk consumed in Korea, 2021.

Commercial Milk Products			
Product Number	Fat (g/100 mL)	Protein (g/100 mL)	Heating Process (Temperature-Holding Time)
1	3.5	3	Ultra-short time (130~130 °C-2~3 s)
2	4	3	Ultra-short time (125~135 °C-2~3 s)
3	3.6	3	Ultra-short time (120~130 °C-2~3 s)
4	4	3	Ultra-short time (120~130 °C-2~3 s)
5	3.6	3	Ultra-short time (120~130 °C-2~3 s)
6	3.5	3	Ultra-short time (120~130 °C-2~3 s)
7	4	3	Ultra-short time (120~130 °C-2~3 s)
8	1	3	Low temperature low time (63 °C-30 min)
9	3.6	3	Ultra-high temperature (135~150 °C-1~4 s)
10	3.6	3	Ultra-high temperature (135~150 °C-1~4 s)
11	3.4	3	Ultra-high temperature (135~150 °C-1~4 s)

## Data Availability

Data is available upon reasonable request.

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
