# Peer review of "Assessment of Stress Levels in Lactating Cattle: Analyzing Cortisol Residues in Commercial Milk Products in Relation to the Temperature-Humidity Index"

_animals, 2023, doi:10.3390/ani13152407_

Round 1

Reviewer 1 Report

Dear editors,

this paper investigates the relationship between milk cortisol residues in commercial milk products and the temperature-humidity index at time of milk production. The authors measuredmilk cortisol concentrations in commercialmilkproductsbetweenJuly and October 2021.They highlighted that milk cortisolconcentration washigherin theperiodsin whichtemperature-humidityindexwashigherthanothermonths as resultofa more stressfulenvironment. Based on this, evaluatingofmilk cortisol concentration may be anusefulltooltoassesstheimpactofenvironmentalstressorson lactating cattle and improvefood safety.The topic is very interesting and current.The manuscript is well written and structured. The references are specific and relevant.

Summary and Abstract: They are well writtenand recap the information contained in the main text without repetitions.

Key-words:they arepertinent and consistent with the topic.You could add “milk stress marker” and change thekey-word “stress” with “heat stress”,beingthefocusofyourtopic.

Introduction: This section properly shows the state-of-the-art resuming the knowledge about this topic.The aim of the study is clearly expressed but the hypothesis should be clearlypresented. In addition,since the reader might not know it, I think you could argue more aboutthe heat stress. Line 79: please, specify in which speciesthese findings have been found. Line 82: the references13 and 15 are referring to the human donor milk, not dairy milk (cattle). You should specifythat you are talking about human milk. I suggest to add some references about cow milk, heat treatment and shelf life to be more pertinent.

Materials and Methods: They are well structured and joined with attractive figures.This section properly describes the sample sizes, the procedures and statistical tests.However, I have a question:you did not know information about the cows (age, breed, stage and number of lactation), management condition and farmsfrom where the milk proceeded.The analyzed milk could come from many different farms. Then, did you consider that severalvariables (age, diet, state of lactation, type of milking, environment) could affect the serum cortisol levels and in turn the milk hormonal content?The approach of your study issuitablewith the purpose to analyze the cortisol residues in the milk taking into account the food inspection side and the impact on the public health. For a physiologic research that wants to confirm the usefullness of cortisol milk measurement as a tool to assessthe environmental stress and animal welfare, it would be interesting to carry out a similar studywith another experimental design, it means having information about animals and their management, using also raw milk as comparison and limiting the variablesas much as possible. I think this is a limit of your study and all these aspects should be discussed very well. In this context, I suggest to add some morereferences to deepen the cortisol pattern in lactating dairy cows, i.e. related to the machine milking impact:

·      Fazio E, Medica P, Cravana C, Ferlazzo A. Release of β-endorphin, adrenocorticotropic hormone and cortisol in response to machine milking of dairy cows. Vet World. 2015 Mar;8(3):284-9. 

Results: They are well presented and joined with very detailed images,adding information to the main text.

Discussion: this section is logically written.However, it would be suitable to argue a little bit more about the limits of your study referring to the physiologic, paraphysiologic and external variables (age, breed, stage of lactation, diet, environment…)that could affect the blood and milk cortisolconcentrations,presenting your persuasive interpretations. Discussing these limits, I think you could enrich this section adding some pertinent references. Line 226: please correct the wrong word. The reference 20 is the same of the 33. Please, correct them.

Overall the paper is very captivating and, since it adds useful information in thefood inspection and public health field, it deserves to be published after minor revisions.

Author Response

Dear Editor of Animals,

A huge thanks to the reviewers for generously devoting their precious time to review and emphasize the important points in the submitted manuscript. It has been observed that queries provided are very valuable, supportive, and logical. These observations have helped us to enhance the overall quality of the manuscript. Hereby, we have prepared a revised version of the manuscript based on the reviewers’ advice. All the changes have been in blue color for easy tracking. Please find the revised version made in the manuscript to reviewer’s comments. We hope that the revised manuscript can comply with the editors and the reviewers’ opinions. We hope our modifications meet the requirements standards of Animals. We greatly appreciate your time spending to assist us to improve our manuscript and for your cooperation.

Sincerely,

Comments and Suggestions for Authors:

Reviewer 1:

This paper investigates the relationship between milk cortisol residues in commercial milk products and the temperature-humidity index at time of milk production. The authors measured milk cortisol concentrations in commercial milk products between July and October 2021.They highlighted that milk cortisol concentration was higher in the periods in which temperature-humidity index was higher than other months as result of a more stressful environment. Based on this, evaluating of milk cortisol concentration may be an usefull tool to assess the impact of environmental stressors on lactating cattle and improve food safety.The topic is very interesting and current.The manuscript is well written and structured. The references are specific and relevant.

Response: The authors appreciate your positive assessment of the manuscript and your valuable comments and logical inquiries regarding its content. Kindly refer to the revised manuscript, where all changes have been indicated in blue color.

Summary and Abstract: They are well writtenand recap the information contained in the main text without repetitions.

Response: Thank you very much for your positive comments on our manuscript.

Key-words: they are pertinent and consistent with the topic.You could add “milk stress marker” and change the key-word “stress” with “heat stress”,being the focus of your topic.

Response: The corrections have been made as per suggestion. Thank you very much. Please refer to the revised text line 42.

Introduction: This section properly shows the state-of-the-art resuming the knowledge about this topic.The aim of the study is clearly expressed but the hypothesis should be clearly presented. In addition, since the reader might not know it, I think you could argue more about the heat stress. Line 79: please, specify in which species these findings have been found. Line 82: the references 13 and 15 are referring to the human donor milk, not dairy milk (cattle). You should specify that you are talking about human milk. I suggest to add some references about cow milk, heat treatment and shelf life to be more pertinent.

Response: Thank you very much for your logical comments. We have accepted and provided responses in the text of the revised manuscript;

The aim of the study is clearly expressed but the hypothesis should be clearly presented.

Response: The combined effects of high temperature and humidity as environmental factors may exacerbate stress in high-producing dairy cattle, resulting in high cortisol concentra-tion in milk. Thus, we hypothesized that in high temperature-humidity index (THI), the level of MCC will also be high. Please refer to the revised text lines 85- 88.

In addition, since the reader might not know it, I think you could argue more about the heat stress.

Response: Heat stress occurs when an animal’s heat load is greater than its ability to dissipate heat. Please refer to the revised text lines 88-89.

Line 79: please, specify in which species these findings have been found.

Response: in dairy cattle [13,14] and human [15,16]. Please refer to the revised text lines 81.

Line 82: the references 13 and 15 are referring to the human donor milk, not dairy milk (cattle). You should specify that you are talking about human milk. I suggest to add some references about cow milk, heat treatment and shelf life to be more pertinent.

Response: :…… in dairy cattle [13,14] and human [15,16].. …… products [17-19]. Please refer to the revised text lines 81,84, 424-425, 432-434. Thank you very much.

Materials and Methods: They are well structured and joined with attractive figures.This section properly describes the sample sizes, the procedures and statistical tests. However, I have a question: you did not know information about the cows (age, breed, stage and number of lactation), management condition and farms from where the milk proceeded.The analyzed milk could come from many different farms. Then, did you consider that several variables (age, diet, state of lactation, type of milking, environment) could affect the serum cortisol levels and in turn the milk hormonal content? The approach of your study is suitable with the purpose to analyze the cortisol residues in the milk taking into account the food inspection side and the impact on the public health. For a physiologic research that wants to confirm the usefullness of cortisol milk measurement as a tool to assess the environmental stress and animal welfare, it would be interesting to carry out a similar study with another experimental design, it means having information about animals and their management, using also raw milk as comparison and limiting the variablesas much as possible. I think this is a limit of your study and all these aspects should be discussed very well. In this context, I suggest to add some more references to deepen the cortisol pattern in lactating dairy cows, i.e. related to the machine milking impact: Fazio E, Medica P, Cravana C, Ferlazzo A. Release of β-endorphin, adrenocorticotropic hormone and cortisol in response to machine milking of dairy cows. Vet World. 2015 Mar;8(3):284-9.

Response: Thank you very much for the positive comments on this section.

However, I have a question:you did not know information about the cows (age, breed, stage and number of lactation), management condition and farms from where the milk proceeded.The analyzed milk could come from many different farms. Then, did you consider that several variables (age, diet, state of lactation, type of milking, environment) could affect the serum cortisol levels and in turn the milk hormonal content?

Response: We highly appreciate your thoughtful inquiries. Because of the comprehensive data concerning the milk collection methodologies employed by the companies and the management practices at the farms are regrettably unavailable. We mentioned this limitation in the last paragraph of discussion section of manuscript lines 329-331 as “Level of cortisol is influenced by not only extreme environmental temperature and humidity, but also other parameters such as feeding practices, housing systems, and methods of milk harvest”.   

The approach of your study is suitable with the purpose to analyze the cortisol residues in the milk taking into account the food inspection side and the impact on the public health. For a physiologic research that wants to confirm the usefullness of cortisol milk measurement as a tool to assess the environmental stress and animal welfare, it would be interesting to carry out a similar study with another experimental design, it means having information about animals and their management, using also raw milk as comparison and limiting the variablesas much as possible. I think this is a limit of your study and all these aspects should be discussed very well. In this context, I suggest to add some more references to deepen the cortisol pattern in lactating dairy cows, i.e. related to the machine milking impact: Fazio E, Medica P, Cravana C, Ferlazzo A. Release of β-endorphin, adrenocorticotropic hormone and cortisol in response to machine milking of dairy cows. Vet World. 2015 Mar;8(3):284-9.

Response: Though individual cow or farm information (e.g., number of milking cows, their man-agement condition, and the intrinsic and extrinsic parameters) may influence MCC, these factors were attenuated as commercial milk is collected from a few number of farms in the same area and mixed altogether, thus, the possible significant factor for the release of MCC is environment, particularly THI in this study. Thank you very much for your suggestion. Thank you very much. Please refer to the revised text lines 105-210.

We mentioned a future research on risk of consuming high level of cortisol in milk products and public health in the last paragraph of discussion section text line 354-357.

Fazio et al., 2015 as a reference is added per your kind suggestion, thank you very much. It was intresting reseach that milk collection method changed cortisol levels, but there was not link to the activation of HPA. For instance, Fazio et al. [11], explained that the activation of hypothalam-ic-pituitary-adrenal axis does not occur in response to machine milking. However, they confirmed that the increase in cortisol levels during milking under normal conditions ap-pears to be regulated by mechanisms other than stress. please refer to the revised text lines 71, 331-334.

Results: They are well presented and joined with very detailed images, adding information to the main text.

Response: The authors are very gratful to you for the kind comments.

Discussion: this section is logically written. However, it would be suitable to argue a little bit more about the limits of your study referring to the physiologic, paraphysiologic and external variables (age, breed, stage of lactation, diet, environment…) that could affect the blood and milk cortisol concentrations, presenting your persuasive interpretations. Discussing these limits, I think you could enrich this section adding some pertinent references. Line 226: please correct the wrong word. The reference 20 is the same of the 33. Please, correct them.

Response: Thank you very much for the positive comments on this section and for your helpful comments.

However, it would be suitable to argue a little bit more about the limits of your study referring to the physiologic, paraphysiologic and external variables (age, breed, stage of lactation, diet, environment…) that could affect the blood and milk cortisol concentrations, presenting your persuasive interpretations. Discussing these limits, I think you could enrich this section adding some pertinent references.

Response: Thank you very much. Please refer to the revised text lines 105-210.

Line 226: please correct the wrong word.

Response: Our mistake has been corrected. Thank you. Please refer to the revised text line 240.

The reference 20 is the same of the 33. Please, correct them.

Response: It has been corrected. Thank you very much.

Overall the paper is very captivating and, since it adds useful information in the food inspection and public health field, it deserves to be published after minor revisions.

Response: Your positive feedback and encouraging words mean a lot to us, and we are delighted to know that you found our work captivating and valuable to the field. We wholeheartedly agree with your assessment. Your suggestion for minor revisions is duly noted, and we assure you that we will carefully consider your feedback to enhance the manuscript further. Thank you very much.

Once again, we would like to express our gratitude to the reviewers for the keen eyes and the comments that were highly valuable and could improve the quality of our presentation. Thank you indeed.

Reviewer 2 Report

This written paper described the relationship between MCC and THI. Before acceptance, some minor issues have to be revised.

Minor revision:

Line 105: Did the authors prepare positive control or negative control in this experiment? A positive control (e.g. blood that would contain cortisol) and a negative control (e.g. the lower layer in a MULTIPLE EXTRACTION step) must be shown to indicate that this measurement method is correct. If the authors did not prepare these controls, this limitation should be written in the discussion.

Line 158-162: Not sure what was included in fixed effects and randam effects.

Author Response

Response to Reviewers

Dear Editor of Animals,

A huge thanks to the reviewers for generously devoting their precious time to review and emphasize the important points in the submitted manuscript. It has been observed that queries provided are very valuable, supportive, and logical. These observations have helped us to enhance the overall quality of the manuscript. Hereby, we have prepared a revised version of the manuscript based on the reviewers’ advice. All the changes have been in blue color for easy tracking. Please find the revised version made in the manuscript to reviewer’s comments. We hope that the revised manuscript can comply with the editors and the reviewers’ opinions. We hope our modifications meet the requirements standards of Animals. We greatly appreciate your time spending to assist us to improve our manuscript and for your cooperation.

Sincerely,

Comments and Suggestions for Authors:

Reviewer 2:

This written paper described the relationship between MCC and THI. Before acceptance, some minor issues have to be revised.

Response: Your positive feedback and encouraging words mean a lot to us, and we are delighted to know that you found our work captivating and valuable to the field. We wholeheartedly agree with your assessment. Your suggestion for minor revisions is duly noted, and we assure you that we will carefully consider your feedback to enhance the manuscript further. Thank you very much.

Minor revision:

Line 105: Did the authors prepare positive control or negative control in this experiment? A positive control (e.g. blood that would contain cortisol) and a negative control (e.g. the lower layer in a MULTIPLE EXTRACTION step) must be shown to indicate that this measurement method is correct. If the authors did not prepare these controls, this limitation should be written in the discussion.

Response: Thank you so much for your detailed comment.
In this experiment, positive and negative matrix controls were not performed during cortisol extraction from commercial milk samples. This limitation should be taken into account for extraction efficiency. We have mentioned in discussion section per your suggestion. Please refer to the revised text lines 346-348.

Line 158-162: Not sure what was included in fixed effects and randam effects.

Response: In this experiment the commercial milk products considered a random effect. The THI from each month of milk production month was treated as a fixed effect. Thank you so much for your detailed comment. Please refer to the revised text lines 173-175.

Once again, we would like to express our gratitude to the reviewers for the keen eyes and the comments that were highly valuable and could improve the quality of our presentation. Thank you indeed.
